# Atmospheric Radiation Measurement (ARM) airborne field campaign data products between 2013 and 2018

Fan Mei[1], Jennifer M. Comstock[1], Mikhail S. Pekour[1], Jerome D. Fast[1], Krista L. Gaustad[1], Beat Schmid[1], Shuaiqi Tang[1,2], Damao Zhang[1], John E. Shilling[1], Jason M. Tomlinson[1], Adam C. Varble[1], Jian Wang[3], L. Ruby Leung[1], Lawrence Kleinman[4], Scot Martin[5], Sebastien C. Biraud[6], Brian D. Ermold[1], Kenneth W. Burk[1]

[1]Pacific Northwest National Laboratory, Richland, WA, 99352, USA
[2]Nanjing University, Nanjing, Jiangsu, 210089, China
[3]Washington University in St. Louis, St. Louis, MO, 63130, USA
[4]Brookhaven National Laboratory, Upton, NY, 11973, USA
[5]Harvard University, Cambridge, MA, 02138, USA
[6]Lawrence Berkeley National Laboratory, Berkeley, CA, 94720, USA

*Correspondence to*: Fan Mei (fan.mei@pnnl.gov)

**Abstract.** Airborne measurements are pivotal for providing detailed, spatiotemporally resolved information about atmospheric parameters, aerosol and cloud properties, thereby enhancing our understanding of dynamic atmospheric processes. For 30 years, the U.S. Department of Energy (DOE) Office of Science supported an instrumented Gulfstream-1 (G-1) aircraft for atmospheric field campaigns. Data from the final decade of G-1 operations were archived by the Atmospheric Radiation Measurement (ARM) user facility Data Center and made publicly available at no cost to all registered users. To ensure a consistent data format and to improve the accessibility of the ARM airborne data, an integrated dataset was recently developed covering the final six years of G-1 operations (2013 to 2018, DOI:10.5439/1999133). The integrated dataset includes data collected from 236 flights (766.4 hours), which covered the Arctic, the U.S. Southern Great Plains (SGP), the U.S. West Coast, the Eastern North Atlantic (ENA), the Amazon Basin in Brazil, and the Sierras de Córdoba range in Argentina. These comprehensive data streams provide much-needed insight into spatiotemporal variability of thermodynamic quantities, aerosol and cloud states and properties for addressing essential science questions in Earth system process studies. This manuscript describes the DOE ARM merged G-1 datasets, including information on the acquisition, data collection challenges and future potentials, and quality control processes. It further illustrates the usage of this merged dataset to evaluate the Energy Exascale Earth System Model (E3SM) with the Earth System Model Aerosol-Cloud Diagnostics (ESMAC Diags) package.

## 1. Introduction

The Earth's climate is changing due to human activities such as fossil fuel burning and deforestation. Atmospheric research is critical for understanding the causes and effects of climate change and for developing strategies to mitigate its impacts (IPCC 2021). Airborne measurements provide a unique observational perspective within the broad objective of understanding

atmospheric processes relevant to climate change. Aircraft are often the only platform from which in-situ observations of chemical and physical parameters through the depth of the troposphere can be obtained; they allow one to follow chemical and

physical processes through the use of pseudo-Lagrangian flight patterns; spatial coverage is usually greater than obtained from surface sites; and observations have a longer time duration than can be obtained from a (non-geostationary) satellite overpass. Vertical profiles can be measured, allowing researchers to obtain vertical profiles of atmospheric states, aerosols, clouds, and trace gas parameters with accuracy that is not attainable with remote sensing. Such parameters are essential for understanding atmospheric physical and chemical processes. Compared to ground observatories, airborne measurements can provide greater

spatiotemporal characterization in a limited time, contributing to a critical, otherwise missing context. This capability is particularly useful for studying rapidly evolving atmospheric phenomena like wildfires, hurricanes, and dust storms. Hence, airborne measurements are a valuable tool in atmospheric studies as it offers several advantages over ground- or satellite-based measurements. (Wendisch et al., 2013; Schumann et al., 2013; Petzold et al., 2013; McQuaid et al., 2013; McFarquhar et al., 2011; Krämer et al., 2013; Brenguier et al., 2013)

The U.S. Department of Energy's Atmospheric Radiation Measurement (ARM) User Facility (Mather and Voyles, 2013)has provided long-term measurements of atmospheric properties by operating ground-based observatories (fixed and mobile)  as well as aerial facilities. Established in 2006, the ARM Aerial Facility (AAF) has led 14 field campaigns using state-of-the-art instruments onboard numerous aircraft (Schmid et al., 2014). Initially, the AAF data was archived individually for each instrument in the International Consortium for Atmospheric Research on Transport and Transformation (ICARTT) file format

(Thornhill et al., 2011), which was developed to fulfill the data management need in 2004. A major strength of the ICARTT file format is its easy-to-use and standard approach to sharing airborne datasets to facilitate broad collaborative scientific research among airborne observation, atmospheric modeling, and satellite observation communities. However, the shortcoming of the ICARTT file format has also long been identified as it is not as efficient as binary formats for data collection and storage and not suitable for extensive multi-dimensional data (https://www.earthdata.nasa.gov/esdis/esco/standards-and-

practices/icartt-file-format). ARM has started efforts to convert the historical field campaign data into Network Common Data Form (NetCDF) (McCord and Voyles, 2016; Rew et al., 1993) – a widely used self-describing data format that supports creating, accessing and sharing array-oriented scientific data.

Additionally, researchers commonly use data analysis software (e.g., Python) that includes functions and library packages that make working with netCDF files easier than text-like files. Using the NetCDF format, various airborne measurements can be

easily combined to generate a merged dataset, which relieves the end-users of the burden of combining data from different data sources. The merged dataset can also aid the research community that uses the abundant ARM aerial data obtained from different field campaigns for diverse science objectives, as detailed in Table 1.

The demand for airborne observations continues to increase with the increase in weather and climate model complexity, as well as the increasing interest in small-scale physical and chemistry processes. Such observations are needed to assess and

validate process-level understanding. To provide a comprehensive view of atmospheric properties, it is desirable to integrate different types of data into a single file, which provides efficient data access for researchers to study the interactions among

aerosol, cloud, and trace gas under various atmospheric conditions in order to understand effects on atmospheric processes and climate. Having all data in one file simplifies data management and reduces the number of files that need to be stored, shared, and accessed. It also minimizes the chances of data loss or errors during file transfers. In a single file, all measured variables are mapped into the same timestamp (e.g., 1 Hz for this study). More details are discussed in session 4. These "merge files" are developed to assist researchers in performing more complex analyses, such as studying the relationships between different types of atmospheric data and carrying out more comprehensive studies with larger datasets. In addition, the merged data, hopefully, encourages collaborations between experimentalists and modelers to combine their expertise and resources to obtain a more complete understanding of the atmospheric phenomena. Thus, after standardizing the AAF data into a NetCDF file format for each field campaign, we used the ARM Data Integrator (ADI, https://github.com/ARM-DOE/ADI) to retrieve and prepare data from each measurement and integrate them into a merged dataset (Gaustad et al., 2014).

This study provides an overview of the airborne datasets collected during seven field campaigns (listed in Table 1) between 2013 and 2018, an introduction to the integrated datasets, and a guide for users to access these datasets on the ARM data archive. Although airborne field campaigns can lead to highly significant scientific findings, they often have a lengthy timeline for generating research papers. Meanwhile, due to pressure to publish quickly and the limited length of funding cycles, researchers do not have enough time to fully leverage field campaign data. In this study, one of the objectives is to draw researchers' attention to these valuable field data, encouraging them to revisit underutilized datasets to reveal new insights. In Section 2, this manuscript provides an overview of the objectives, flight information, and measurements of the seven field campaigns AAF carried out with the G- aircraft between 2013-2018. Section 3 has three subsections: data quality is discussed by comparing the in situ measurements against other measurements; then, we briefly outline how to use the merged dataset to evaluate aerosol-cloud interactions represented in Earth system models; in the 3$^{rd}$ subsection, we outline data collection challenges, lessons-learned, and also future potentials of these airborne data. Further detail on the data structure is given in section 4. In addition, a summary section describes the potential of this merged dataset after explaining the data file structure and availability of the data.

## 2.  Campaign objectives and flight patterns

### 2.1.  ARM airborne facility supported field campaigns between 2013 and 2018.

The ARM program is a U.S. Department of Energy scientific user facility that aims to provide observation data to improve our understanding of the Earth's atmosphere and its interactions with land surface and oceans. One of the key components of the ARM program is to conduct periodic field campaigns, which are intensive measurement periods at specific locations focused on specific scientific questions. These field campaigns involve deploying state-of-the-art instruments with both ground station and airborne platforms to collect measurements of atmospheric states, radiation, clouds, precipitation, aerosols, and trace gas variables, often including collaborations with other research programs and institutions. Table 1 shows the campaign locations, flight hours, and the scientific objectives of each field study carried out by the AAF between 2013 and 2018. The word cloud

depicted in Figure S1 encapsulates the content of all AAF-supported field campaign publications, showing that the AAF has played a pivotal role in advancing research across various important atmospheric domains. The visual representation highlights a concentration of terms related to key atmospheric topics, including aerosols, clouds, precipitation properties, and the intricate processes governing their interactions. This observation underscores the multifaceted support provided by the AAF to the scientific community, contributing to the exploration and understanding of crucial atmospheric phenomena.

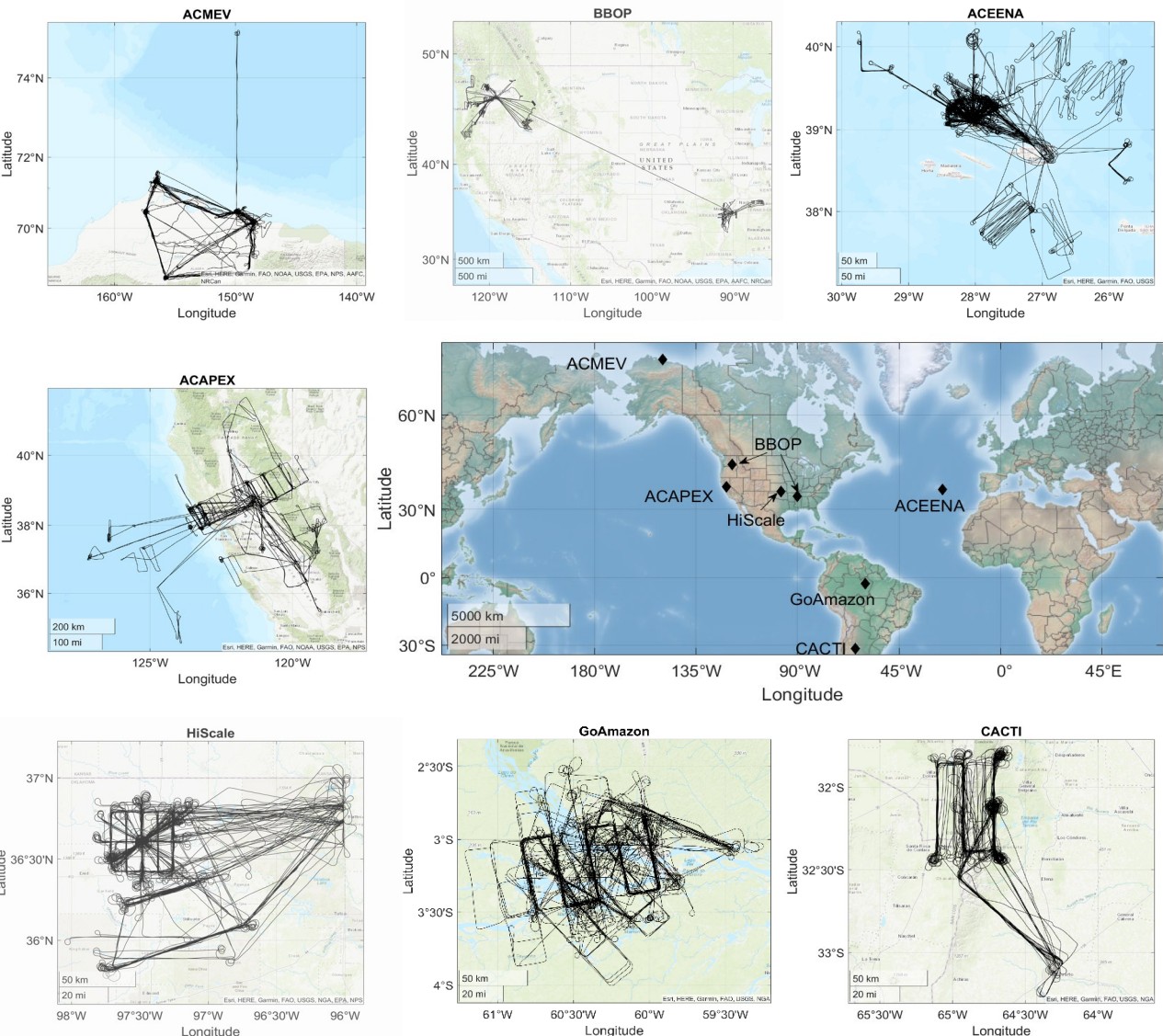

**Figure 1.** Flight tracks overlaid with the campaign location map from seven AAF field campaigns between 2013-2018 using MATLAB$^{@}$

Table 1. A brief summary of the seven AAF field campaigns between 2013-2018.

| AAF-supported field campaigns (ARM location)* | Dates, Number of flights (flight hours) | Scientific objectives (Campaign websites) |
|---|---|---|
| Biomass Burning Observation Project (BBOP), 2013 (OSC: Pasco, WA, USA and Memphis, TN, USA) | 1 July – 24 Oct. 2013, 35 (97.7 hrs.) | Campaign website: https://arm.gov/research/campaigns/aaf2013bbop<br>The BBOP field campaign aimed at improving understanding of the physical and chemical processes of biomass burning aerosol properties. Aircraft-based measurements were used to study the properties of biomass-burning aerosols between the fire and regions several hours downwind. The observations quantified the time evolution of the aerosols' properties, such as their microphysical, morphological, chemical, hygroscopic, and optical characteristics. The goal was to use the data to constrain processes and parameterizations in a Lagrangian model of aerosol evolution and better understand the radiative effects of biomass burning (Collier et al., 2016; Zhang et al., 2018; Kleinman et al., 2020). |
| Observations and Modeling of the Green Ocean Amazon (GoAmazon), 2014 (MAO: the Manacapuru region of the Brazilian Amazon) | 15 Feb. – 15 Oct. 2014 35 (89.5 hrs.) | Campaign website: https://arm.gov/research/campaigns/amf2014goamazon<br>The GoAmazon experiment aimed to study how aerosols and surface fluxes influence cloud properties and how pollutant outflow from a tropical megacity affects aerosol and cloud life cycles. The data collected during the experiment helped improve tropical rainforest models and better understand the chemical and physical processes of anthropogenic-biogenic interactions that affect the production of secondary organic aerosols (SOA). The experiment sought to answer several questions, including the effects of urban emissions on SOA production, the absence of new particle formation in the pristine Amazon, and the influence of the Manaus pollution plume on cloud condensation nuclei (CCN) activities and aerosol optical properties. Additionally, the experiment helped to understand how biogenic volatile organic compound (BVOC) emissions impact HOx chemistry in the unpolluted Amazon environment, how they are changing, and how anthropogenic emissions modify the impact of BVOC emissions on HOx chemistry in the Amazon (Wang et al., 2016; Martin et al., 2017; Fan et al., 2018). |
| ARM Cloud Aerosol Precipitation Experiment (ACAPEX), 2015 | 14 Jan. – 12 Mar. 2015 29 (106.3 hrs.) | Campaign website: https://arm.gov/research/campaigns/amf2015acapex<br>The overarching objectives of ACAPEX and concurrent CalWater 2 projects were to provide measurements and improve understanding of several key aspects of atmospheric science. These included documenting and quantifying the |

| | | |
|---|---|---|
| (ACX: in coastal CA, USA) | | structure, evolution, and moisture budgets of the atmospheric rivers (ARs), improving understanding and modeling of the influence of the tropics on extratropical storms and ARs, characterizing aerosols and their microphysical properties over the Pacific Ocean, and understanding aerosol-cloud-precipitation interactions in clouds transitioning from maritime to orographic regime. The projects aimed to answer specific questions related to the evolution and structure of ARs and associated clouds and precipitation, the role of tropical convection and ocean mixed-layer processes in AR evolution, the critical dynamical processes that modulate cloud and precipitation from landfalling ARs, and the influence of aerosols on precipitation and cyclogenesis. The projects also aimed to understand the frequency and characteristics of aerosol transport across the Pacific and their influence on cloud and precipitation in both AR and non-AR conditions (Thompson et al., 2016; Lacher et al., 2018; Levin et al., 2019). |
| Airborne Carbon Measurements (ACME V), 2015 (NSA: North Slope of Alaska, Alaska, USA ) | 1 June – 15 Sep. 2015 38 (139.0 hrs.) | Campaign website: https://arm.gov/research/campaigns/aaf2015armacmev The ARM ACME V campaign aimed to collect trace gas and atmospheric properties over the North Slope of Alaska to address multiple science objectives to enhance the understanding of Earth's weather patterns and reduce uncertainty in global and regional climate simulations and projections. The campaign aimed to measure and model the exchange of carbon dioxide, water vapor, and other trace gases, develop and test measurement and modeling approaches to estimate regional carbon balances and human-made sources, characterize atmospheric mixing ratios, aerosol and cloud properties, and upwelling and downwelling radiation budgets, evaluate interactions between aerosols and clouds, and relate spatial and seasonal differences in greenhouse gas sources and atmospheric transport to variations in $CO_2$ and $CH_4$ mixing ratios (Maahn et al., 2017; Creamean et al., 2018; Tadić et al., 2021). |
| Holistic Interactions of Shallow Clouds, Aerosols, and Land-Ecosystems (HI-SCALE), 2016 (SGP: Oklahoma, USA.) | 24 April – 23 Sep. 2016 38 (106.6 hrs.) | Campaign website: https://arm.gov/research/campaigns/sgp2016hiscale The scientific issues addressed by the HI-SCALE related to understanding the processes that control the formation and properties of shallow convective cumulus clouds. One of the key factors affecting shallow cloud formation was the heterogeneity of land use, vegetation, and soil moisture conditions. Other essential factors included cloud population size, organization, and entrainment mixing, as well as the properties of aerosols, such as their size, number concentration, composition, and mixing state. To address these issues, scientists investigated how variations in vegetation, soil moisture, surface albedo, and |

| | | |
|---|---|---|
| | | downwelling radiation affect surface heat fluxes and the sub-grid variability of temperature, humidity, and vertical mixing in the atmospheric boundary layer. They were also studying the impact of entrainment mixing at the boundary layer top on cloud-aerosol interactions and CCN concentrations, as well as the contribution of new particle formation, secondary organic aerosol formation, and aerosol growth to CCN concentration. Scientists used large eddy simulation modeling to capture the observed temporal and spatial variability of surface fluxes, boundary layer mixing, aerosol and CCN properties, cloud-aerosol interactions, and cloud properties over the SGP site to better understand the relative impacts of different aerosol sources. Ultimately, they hoped to use high-resolution aircraft data, coupled with large eddy simulation modeling and routine ARM measurements, to develop new parameterizations of sub-grid scale variability associated with boundary layer turbulence and shallow clouds (Fast et al., 2019b; Fast et al., 2022; O'Donnell et al., 2023). |
| Aerosol and Cloud Experiments in the Eastern North Atlantic (ACE-ENA), 2017/8 (ENA: Azores, Portugal.) | 15 June 2017 – 28 Feb. 2018 39 (151.9 hrs.) | Campaign website: https://arm.gov/research/campaigns/aaf2017ace-ena The main objective of the ACE-ENA study was to investigate the fundamental processes that control the properties and interactions of aerosols and clouds in different meteorological and cloud conditions in Northern Atlantics. The study also aimed to provide high-quality in situ measurements to improve ground-based retrieval algorithms at the ENA site, enabling better use of routine measurements for model evaluation. The scientific questions and objectives were organized into five themes. The first theme is the budget of Marine Boundary Layer (MBL) Cloud Condensation Nuclei (CCN) and its seasonal variation. The second theme focused on the effects of aerosols on clouds and precipitation, including how ground-based lidar and CCN measurements can better infer CCN concentration at the cloud base and how various CCN concentrations affect cloud microphysics and precipitation potential. The third theme addressed cloud microphysical and macrophysical structures and entrainment mixing, including the mesoscale variabilities of cloud microphysics, the thermodynamic and spatial characteristics of cold pools, and the relationships between the entrainment rate and microphysical effects. The fourth theme was advancing retrievals of turbulence, cloud, and drizzle, which includes validating and quantifying the uncertainties in turbulence, cloud, and drizzle microphysical properties obtained from vertically pointing observations and improving 3D cloud and drizzle retrievals from scanning radars. Finally, the |

| | | fifth theme was model evaluation and process studies, including comparing predictions of global models using "nudged" or "specified" meteorology with airborne observations and examining the CCN budget terms and processes driving the vertical structure and mesoscale variation of aerosol, cloud, and drizzle fields using validated/constrained General Circulation Models (GCM) and Large Eddy Simulation (LES) models (Zawadowicz et al., 2021; Wang et al., 2022; Zhang et al., 2023b). |
|---|---|---|
| Cloud, Aerosol, and Complex Terrain Interactions (CACTI), 2018 (COR: Córdoba, Argentina.) | 1 Nov. – 15 Dec. 2018 22 (75.4 hrs.) | Campaign website: https://arm.gov/research/campaigns/amf2018cacti CACTI aimed to improve the understanding of factors governing the life cycles of orographic convective clouds in a global hotspot for the development of such clouds where few in situ measurements had been previously collected. The campaign focused on two categories: relatively shallow cumulus and stratocumulus clouds and deep convective clouds. Specific objectives for shallow clouds included understanding how the boundary layer flows and the lower free troposphere combined to control cloud evolution and how clouds modified and mixed boundary layer moisture and aerosols into the free troposphere. Specific objectives for deep convective clouds included isolation of the effects of various environmental factors on the initiation, growth, and organization of these clouds and assessing how they, in turn, affect soil moisture, surface fluxes, and aerosol properties. Studies using CACTI datasets continue to be used to advance understanding of interactions between convective clouds and their surrounding environment, including how aerosol and cloud properties affect one another, information that is being used to evaluate and improve weather and climate models (Veals et al., 2022; Varble et al., 2021). |

\* ARM site identifier was included with each field campaign, such as OSC for the BBOP campaign.

## 2.2. Description of the airborne platform and sensors

The ARM Aerial Facility (AAF) took over operations of the G-1 research aircraft in 2010. The G-1 is a medium-sized business aircraft that has been providing atmospheric measurements for various DOE programs from 1989 to 2018  (Figure 2). The aircraft is powered by two turboprop engines, and it has a 1,900 kg cabin payload and an operational range of approximately 4,000 km for ferry flight at high altitudes and 800 km for boundary layer sampling, making it well-suited for deployments to remote locations. The G-1 can continuously fly for 3-4 hrs (5 hours with minimum payload). with a full payload at a cruising speed of approximately 100 m/s up to a maximum altitude of approximately 7.6 km, which allows it to access a wide range of atmospheric conditions and altitudes for measurement purposes.

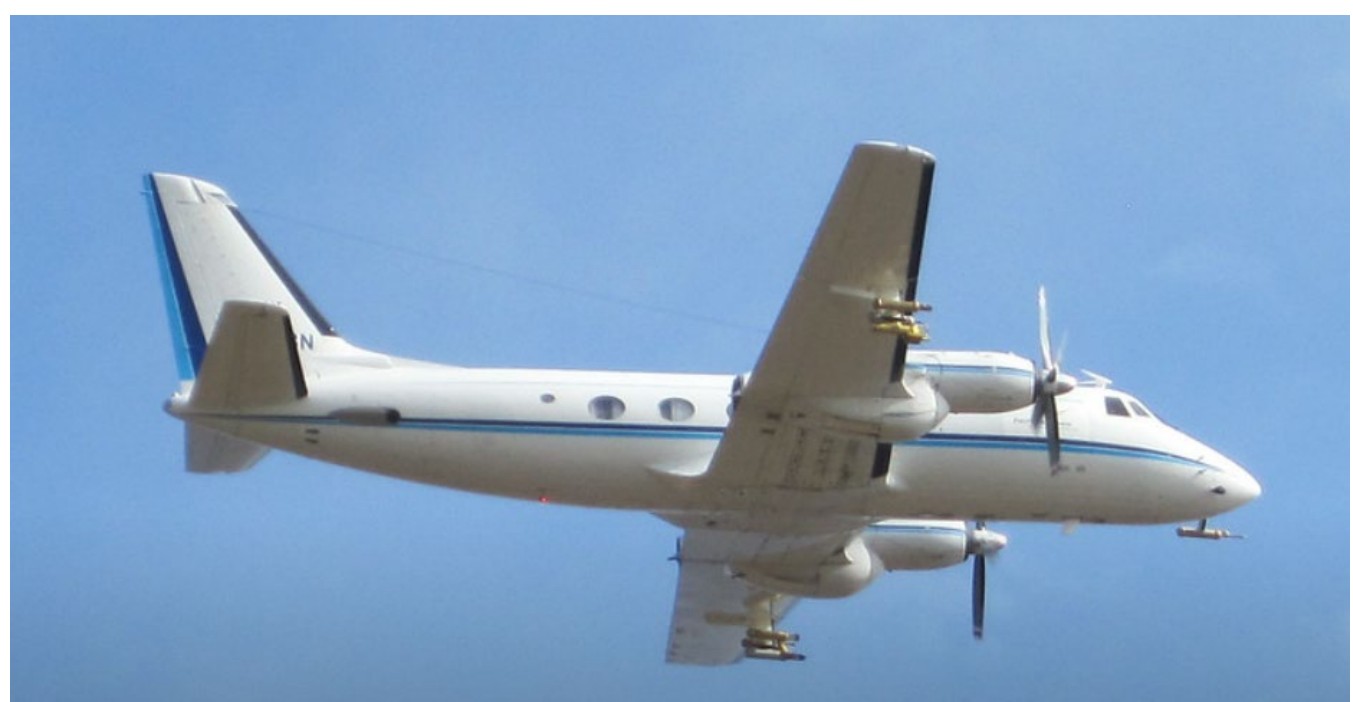

**Figure 2. The G-1 research aircraft flying over the ARM ENA site (Image courtesy of Ed Luke, Brookhaven National Laboratory)**

The ARM G-1 is equipped with a suite of instruments and sensors for making measurements of atmospheric parameters (including temperature, humidity, pressure, winds, and turbulence), radiation, concentrations of atmospheric gases, and types, concentrations, and sizes of aerosol, cloud, and precipitation particles, as shown in Table 2. These instruments and sensors are mounted on the 'aircraft's exterior and inside the cabin. The measurement variables for individual data streams are listed in Table S1, which are used as input data for the merged data product. Note that some measurements, including those related to redundant cloud measurements and radiation, are not currently incorporated into the merged dataset described here due to the complexity of the existing data. However, these measurements remain under consideration for future inclusion, reflecting a commitment to expanding the 'dataset's scope as technology and analytical capabilities evolve. Additionally, photos, videos, and some guest instruments are non-traditional data types included in the ARM individual archive but not with this merged data. Driven by the various campaign objectives and payload limitations, the measurements available for each field campaign are listed in Table S2. The data collected by these instruments are used to improve our understanding of the 'Earth's climate system and to support the development of climate models and other tools for studying and predicting atmospheric and climate phenomena (Mei et al., 2020).

Table 2. The ARM G-1 payload instruments that were included in the merged data product.

## Atmospheric State and Aircraft State

| Instrument | Description | Source/Supplier |
|---|---|---|
| GPS (Global Positioning System) DSM 232 GPS/INS | Position and velocity | Trimble |
| VectorNav INS | Position and velocity | VINS |
| Miniature Integrated GPS/INS Tactical System (C-MIGITS) III | Inertial navigation system/global positioning system: position, velocity, attitude | Systron Donner |
| Aircraft Integrated Meteorological Measurement System - 20 (AIMMS-20) | 5-port air motion sensing: true airspeed, angle-of-attack, side-slip<br>Meteorology: temperature, relative humidity, and pressure.<br>Inertial navigation system/global positioning system: position, velocity, attitude | Aventech |
| Chilled Mirror Hygrometer – General Eastern 1011C | Dewpoint temperature | General Eastern |
| Rosemount 1201F1 | Pressure | Goodrich Corporation |
| Rosemount E102AL/510BF | Temperature | Goodrich Corporation |

## Cloud Properties

| Instrument | Description | Source/Supplier |
|---|---|---|
| Fast-Cloud Droplet Probe (FCDP) | Cloud particle size distribution 2 to 50 μm | Stratton Park Engineering Company |
| 2-Dimensional Stereo Probe (2D-S) | Cloud particle size distribution 10 to 3,000 μm | Stratton Park Engineering Company |
| High Volume Precipitation Spectrometer Version 3 (HVPS) | Precipitation particle size distribution 150 to 19,600 μm | Stratton Park Engineering Company |

## Aerosol Properties

| Instrument | Description | Source/Supplier |
|---|---|---|
| Aerosol Isokinetic Inlet | Sample stream of dry aerosol, sizes < 5 microns | Brechtel Manufacturing Inc. |
| Counterflow Virtual Impactor (CVI) Inlet | Sampling of cloud droplet residuals | Brechtel/PNNL Build |
| 3-Wavelength Integrating Nephelometer, Model 3563 | Aerosol scattering coefficient 450, 550, 700 nm | Trust Science Innovation (TSI) Inc. |

| | | |
|---|---|---|
| 3-Wavelength Particle Soot/Absorption Photometer (PSAP) | Aerosol absorption coefficient 462, 523, 648 nm | Radiance Research |
| Ultrafine Condensation Particle Counter (UCPC), Model 3025A | Total aerosol concentration >0.003 µm | Trust Science Innovation (TSI) Inc. |
| Condensation Particle Counter (CPC), Model 3772 | Total aerosol concentration >0.010 µm | Trust Science Innovation (TSI) Inc. |
| Dual-Column Cloud Condensation Nuclei Counter (CCN) | Concentration of cloud condensation nuclei at two specified supersaturations | Droplet Measurement Technologies |
| Passive Cavity Aerosol Spectrometer-100X (PCASP) | Size distribution 0.10 to 3 µm | Particle Measuring Systems Inc. (PMS) |
| Ultra-High Sensitivity Aerosol Spectrometer (UHSAS) | Aerosol size distribution 0.060 to 1 µm | Droplet Measurement Technologies |
| Fast Integrated Mobility Spectrometer (FIMS) | Aerosol size distribution 0.010 to 0.450 µm | BNL Build |
| Single Particle Soot Photometer (SP2) | Soot spectrometry | Droplet Measurement Technologies |
| High-Resolution Time-of-flight Aerosol Mass Spectrometer (HR-ToF-AMS) | Particle chemical composition | Aerodyne Inc. |

## Gas Phase Measurements

| Instrument | Description | Source/Supplier |
|---|---|---|
| N2O/CO -23r | Concentration of CO, N2O, and H2O | Los Gatos |
| SO2 - Model 43i | Concentration of SO2 | Thermo Scientific/BNL |
| O3 - Model 49i | Concentration of O3 | Thermo Scientific |
| Oxides of Nitrogen | Concentration of NO, NO2, and NOy | Air Quality Devices/BNL |


### 3. Data quality evaluation - consistency among observations

The AAF's airborne measurements are considered accurate and reliable because they are obtained directly from the atmosphere using well-calibrated instruments. Numerous prior studies have systematically assessed ARM data quality, employing methods such as laboratory evaluation based on community-accepted standards, comparing similar properties across different
instruments, and conducting intercomparisons across diverse platforms (from ground to airborne or airborne to airborne) (Bond et al., 1999; Lance et al., 2010; Kassianov et al., 2015; Kassianov et al., 2018; Mei et al., 2020; Zawadowicz et al., 2021; Kulkarni et al., 2023; Tang et al., 2023; Zhang et al., 2023b). Table S3 lists AAF measurements and uncertainties of

atmospheric properties, including temperature, humidity, aerosol concentrations, cloud particle sizes, and radiation levels. (Mei et al., 2020)

The AAF airborne data are also often used as a benchmark or standard for other measurements, especially those from remote sensing technologies such as satellites, ground-based radars, and lidars. Junghenn Noyes et al. validated remote sensing retrievals with help from ground-based and airborne measurements. Their study enhanced the understanding of smoke particle behavior and its implications for remote sensing. (Junghenn Noyes et al., 2020) Mech et al. showcased how integrating airborne data into the PAMTRA (Passive and Active Microwave TRAnsfer) validation process enhances the model's skill in accurately

simulating microwave measurements. The detailed comparison between simulated and observed data helps understand the model's performance in real-world conditions, leading to a more robust and reliable tool for atmospheric research. (Mech et al., 2020) Yang et al. developed a new method to estimate supersaturation fluctuations in stratocumulus clouds using ground-based remote-sensing retrievals. Then, they used the airborne data to validate these estimations. (Yang et al., 2019) Wu et al. retrieved profiles of marine boundary layer (MBL) cloud and drizzle microphysical properties from ground-based

observations, validated by aircraft measurements over the Azores. (Wu et al., 2020) Zhang et al. (2023) evaluated cloud droplet number concentrations using multiple ground-based methods validated through aircraft in situ measurements. (Zhang et al., 2023a)

Furthermore, research based on collected aircraft data led to advancements, characterization, and understanding of atmospheric processes. (Martin et al., 2017; Fast et al., 2011; Fast et al., 2019a; Fast et al., 2019b; Varble et al., 2021; Wang et al., 2022)

For example, various studies have utilized aircraft measurements to characterize aerosol and cloud properties while advancing the understanding of aerosol chemistry and cloud microphysical properties and processes, including investigations over the North Atlantic, Amazon basin, and Southern Great Plains. (Shrivastava et al., 2019; Shilling et al., 2018; Zawadowicz et al., 2020; Wang et al., 2023b; Fast et al., 2024) The airborne data has also been used to examine the vertical variability of aerosol properties over the Southern Great Plains, contributing to a better understanding of the distribution and impact of aerosols at

different atmospheric levels. (Wang et al., 2016; Fast et al., 2022)

This study further demonstrates the comparison of AAF data with ground-based remote sensing retrieval. The ARM G-1 aircraft was deployed above or near the ENA and SGP sites during field campaigns like ACE-ENA and HI-SCALE. Various instruments employing diverse observational techniques have measured atmospheric parameters, aerosol, and cloud properties from ground and airborne perspectives. These coordinated deployments enable a thorough assessment of robustness and

statistical representativeness across collocated measurements.

Comparing airborne and ground-based measurements involves evaluating data from two platforms that differ in spatial and temporal resolutions, and measurement techniques. Thus, three potential biases exist in the measurements – spatial, temporal, and instrumental. The instrumental bias is typically due to the differences in sensors, calibration, and data processing techniques between the two platforms. Airborne measurements usually provide in situ spatiotemporal data over leveled flight

legs at different altitudes and can capture data over various, even difficult-to-access, terrains. Meanwhile, ground-based remote sensing data usually provide continuous monitoring at a fixed location with limited spatial coverage or less vertical resolution.

Our efforts focus on minimizing the temporal and spatial biases to ensure accurate and meaningful comparisons. We selected the comparison period by aligning the data acquisition times for both airborne and ground-based measurements as closely as possible. To ensure that both airborne and ground-based measurements are georeferenced accurately. For instance, ground-based remote sensing uses height or the altitude above the ground level (AGL) as the vertical geographic coordinate. In contrast, airborne data usually uses the mean sea level (MSL) altitude, which can be converted to the AGL. We then use interpolation techniques to match the spatial resolutions of airborne and ground-based data.

## 3.1. Airborne data quality control

### 3.1.1. Aircraft integrated meteorological measurement system data

The aircraft platform velocity, position, and attitude were monitored by four redundant sensors: DSM, C-MIGITS, VectorNav, and AIMMS-20, as shown in Table 2. The DSM is the primary choice of the data source for the merged dataset, followed by the C-MIGTS, AIMMS-20, and VectoNav. For static temperature measurement, the uncertainty of the field data is ±0.5 K. The static pressure has a measurement uncertainty of 0.5 hPa. The standard measurement uncertainties were ±2 K for the chilled mirror hygrometer. The AIMMS-20 provides both meteorological measurements and wind vectors (https://aventech.com/products/aimms20.html). The relative humidity sensor has an accuracy of 2 % in the operating range of 0- 100 %. A calibration flight pattern was performed with each installation of the AIMMS-20 in each field campaign, improving the high-resolution wind data accuracy. The AIMMS-20 calibration uses two different procedures. The aerodynamic calibration maneuver helps to determine aerodynamic errors induced by the aircraft itself. Then, the aircraft was operated to complete an inertial system calibration maneuver and capture minor alignment errors (i.e., cross-axis error) between the gyros, accelerometers, GPS antenna baseline and the primary reference frame of the inertial measurement unit (IMU). Combining the measurements from the instruments that make up the AIMMS-20 (the air data probe (ADP), the GPS, and the IMU) provides the wind speed with an accuracy of 0.5 m/s for the North and East components and 0.75 m/s for the vertical wind. The AIMMS-20 wind parameters, such as wind speed and wind direction, were compared with the ARM ground-based Doppler lidar (DL) retrieved wind parameters (Newsom et al., 2019) in Fig. 3. The ARM DL is an active remote-sensing instrument that provides time and range-resolved measurements (Newsom and Krishnamurthy, 2020). The DOE ARM user facility operates several scanning coherent Doppler lidar systems in the near-infrared (1.5 μm) at ARM's ground-based observatories and mobile facilities (Newsom et al., 2017). The Doppler lidar horizontal wind profiles Value-Added Product (DLPROF-WIND VAP, https://www.arm.gov/capabilities/vaps/dlprof-wind) provides accurate height-resolved measurements of wind speed and direction in 15-min resolution (Newsom et al., 2019). Figure 3 compares G-1 aircraft winds to DL winds for level flight legs under cloudless conditions during the ACE-ENA field campaign. The aircraft data were averaged to match the DL time interval when the aircraft location was within a 3 km distance from the ARM ground site. Limited by frequent clouds and the DL data availability, only leveled flight legs from 12 flights between June 26 to July 19, 2017, were included in the comparison. We achieved a reasonably good comparison between the AIMMS-20 and DL wind parameters, especially for the

wind direction. The wind speed plot exhibits increased dispersion, particularly at lower wind speeds (<5 m/s), possibly attributed to the heterogeneous nature of wind speeds at different heights in the Doppler lidar data. Notably, data from a single aircraft flight path may not accurately reflect the values measured by the Doppler lidar. It should be noted that there are fundamental differences in aircraft-based wind measurements (in-situ, spatial averages) and the Doppler Lidar technique (single point, time/height averaged), which never allows perfect agreement between the two data sets.

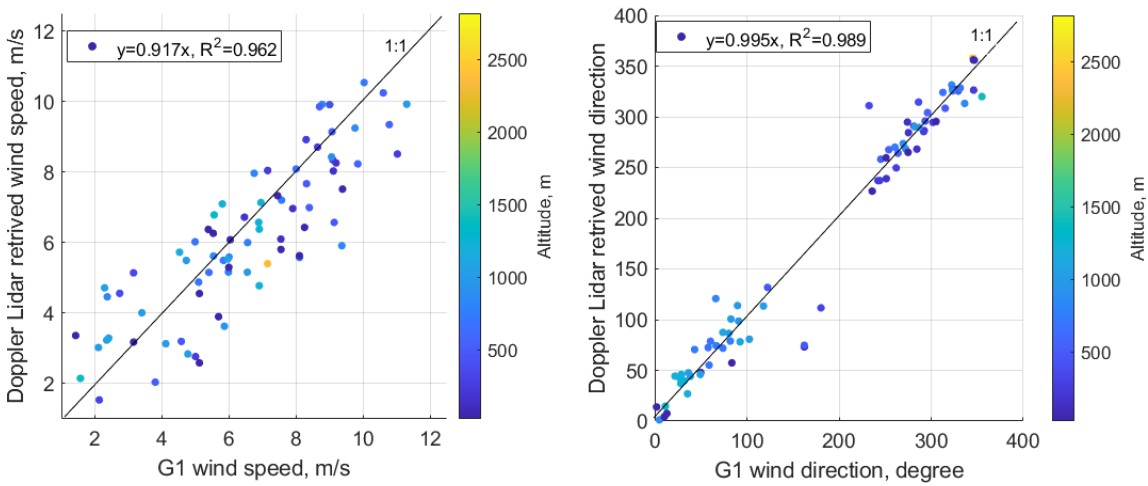

**Figure 3**. Wind parameters comparison between the AIMMS-20 and Doppler Lidar colored by the aircraft altitude for cloudless flight legs during the ACE-ENA campaign.

### 3.1.2. Aerosol payload data

Aerosol measurements aboard the G-1 platform were performed downstream of an isokinetic inlet with a Nafion dryer to ensure relative humidity below 40% for the air sample. The isokinetic aerosol inlet was designed and built by Brechtel Manufacturing Inc. (Hayward, CA) and modified by the Pacific Northwest National Laboratory (PNNL). Previous closure studies have shown that the isokinetic inlet allowed sampling of the aerosols up to 5 μm aerodynamic diameter, and the transmission efficiency is around 50% at 1.5 μm (Kassianov et al., 2015; Kassianov et al., 2018; Mei et al., 2020; Kassianov et al., 2021). The best-estimate aerosol size distribution (BEASD) data product was created by merging aerosol size distribution from several (up to four) aerosol and cloud sensors under dry conditions (<40% RH). Two aerosol spectrometers were used as primary data sources: The Fast Integrating Mobility Spectrometer (FIMS; 10 to 600 nm; 30 log-spaced bins, 16.31 bins per decade) and the Passive Cavity Aerosol Spectrometer (PCASP, 0.095 to 2.9 μm size range). As shown in Fig. 4, we demonstrate a reasonable agreement between the integrated total number concentration from the BEASD and the total number concentration measured by the condensation particle counter (CPC, TSI, model 3772, >10 nm). This CPC was calibrated following the World Calibration Center for Aerosol Physics (WCCAP) guideline. Compared with the lab standard CPC, the

typical uncertainty is ~10% (Mei et al., 2020). Two cloud probes were used as a secondary source of aerosol measurements to cover the super-micron size range: the Cloud Aerosol Spectrometer (CAS, 0.55 to 12.73 μm) and the Fast Cloud Droplet Probe (FCDP, 0.75 to 13.49 μm). Aerosol chemical composition, measured by the Aerodyne High-Resolution Time-of-Flight Aerosol Mass Spectrometer (HR-ToF-AMS), was used to estimate the aerosol refractive index (RI) necessary for correction of the aerosol equivalent optical size into equivalent geometric size (Li et al., 2023; Hand and Kreidenweis, 2002; Freedman et al., 2009).

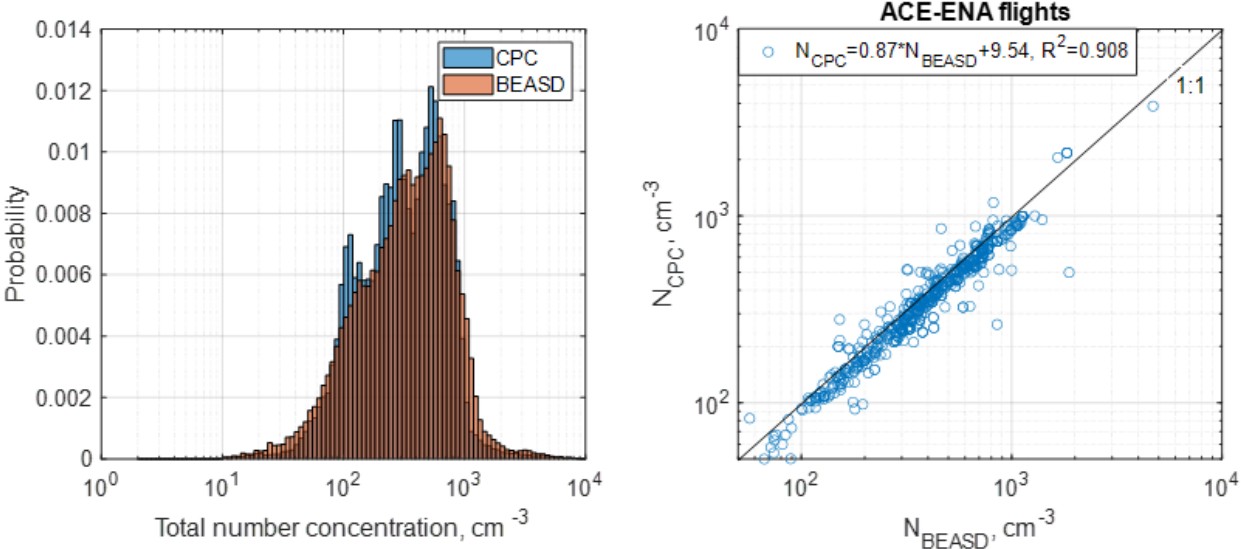

**Figure 4**, Aerosol total number concentration comparison between the G-1 CPC 3772 and the best estimate aerosol size distribution during the ACE-ENA flights.

The scattering and absorption coefficients are two essential parameters for understanding the optical properties of atmospheric aerosols, as they describe the scattering and attenuation of sunlight and the visibility of the atmosphere. The payload of the G-1 incorporated two aerosol optical instruments to measure these properties. A Nephelometer (TSI, model 3563) was used to measure light scattering by aerosol particles at three wavelengths (Uin and Goldberger, 2020). It uses a light source to illuminate a sample of aerosol particles and measure the intensity of the total and backscattered light. The scattering coefficient from a nephelometer is defined as the ratio of the scattered light flux to the incident light flux and is typically expressed in units of inverse megameters (Mm$^{-1}$). A Particle Soot Absorption Photometers (PSAP, Radiance Research) is used to estimate aerosol absorption by measuring the attenuation of a light beam passing through aerosols deposited on a filter (Springston, 2018). After correcting for the filter effect and scattering impacts on the absorption values, the amount of light absorbed by the particles for three wavelengths was recorded (Bond et al., 1999). Under dry conditions, the accuracies for the scattering and absorption coefficients are 25% and 20%, respectively (Rosati et al., 2016). The uncertainties of those airborne

measurements might be even larger due to the complex field conditions. The sum of the absorption and scattering of aerosol particles (i.e., aerosol extinction) provides a measure of the effect of aerosols on radiant energy that passes through the atmosphere.

In addition, the aerosol extinction coefficient can be calculated from aerosol size distribution and chemical composition using Mie theory with certain assumptions (Mie, 1908; Bohren and Huffman, 1998). Here, assuming spherical particles and homogeneous composition in the estimated size range, we estimated the extinction coefficients of the aerosol particles and compared the values with the in-situ measurements from the summation of the values from PSAP and Nephelometer in Fig. 5. The comparison data were only from level flight legs when aerosol sampling was under dry and isokinetic condition. The BEASD estimation and the Nephelometer and PSAP summation in the left statistic plot achieved a reasonably good agreement. The dots on the scatter plot are color-coded by the relative humidity. This scatter plot indicates that the simple assumption, which ignores how the aerosol properties vary with particle sizes, shapes, and refractive indices and also neglects the ambient relative humidity effect, introduces uncertainty on the aerosol properties.

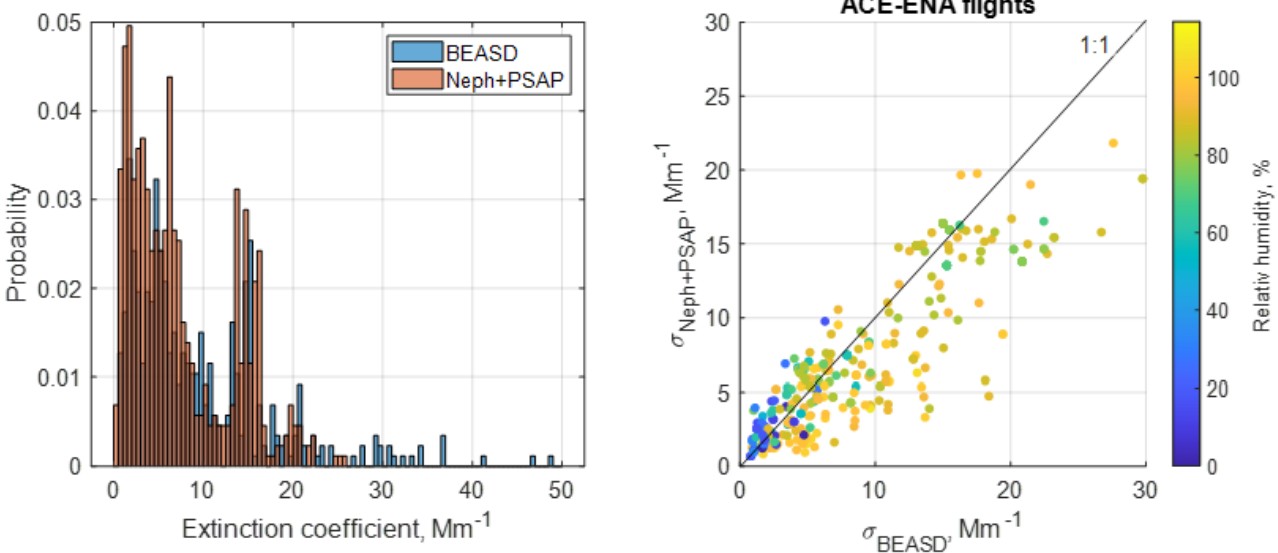

**Figure 5.** Aerosol extinction coefficient comparison between the estimated values based on the BEASD size distribution and the summation of the in-situ measurements from the nephelometer and PSAP during the ACE-ENA flight for the leveled flight legs.

### 3.1.3. Cloud payload data

A variety of cloud optical sensors were deployed during the seven AAF campaigns. The cloud measurement methods can be separated into light scattering (FCDP) and shadow imaging (2D-S and HVPS), covering a wide range from small cloud droplets to precipitation elements. The FCDP measures cloud droplets in the size range of 2-50 μm diameter. Droplets are

detected and sized depending on how much light they scatter in a specific angular range when illuminated by a focused laser

beam (Lance et al., 2010). The 2D-S and HVPS are optical array probes that restore images from passing hydrometeors'

shadows. Using two cameras positioned at different angles to create a stereo image of each cloud particle, 2D-S and HVPS

provide information about the particle's size, shape, and orientation at different size ranges. (Glienke and Mei, 2020, 2019)

Based on previous studies, the undercounting bias of measured droplets between 3 and 20 µm diameter is around 20%, and

for the droplet larger than 20 µm diameter, the uncertainty is up to 50% (Glienke et al., 2023; Mei et al., 2020). In addition, a

merged size distribution based on the FCDP, 2D-S and HVPS was created to cover the size range of cloud elements from 2 to

9075 µm.

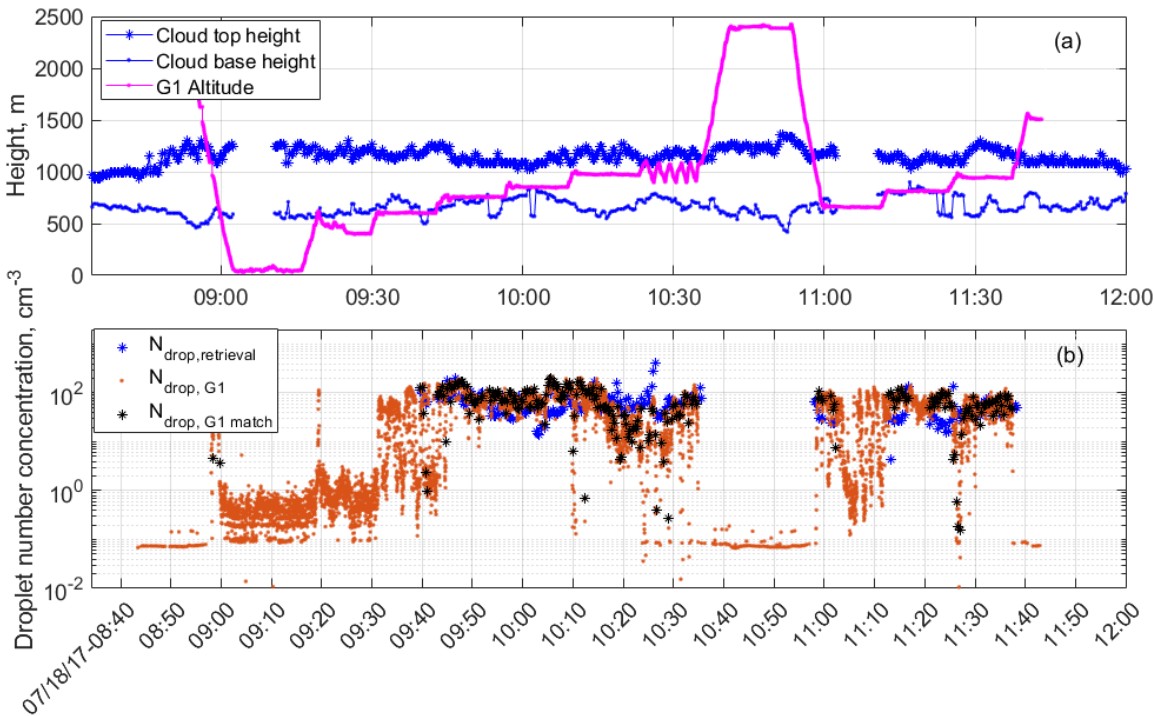

**Figure 6.** Cloud droplet number concentration comparison between the integrated value from the merged cloud droplet size

distribution and the ground-based Raman Lidar retrieval value on July 18, 2017, during the ACE-ENA field campaign.

Fig. 6 (a) demonstrates one case under a stratocumulus cloudy day and shows three lines representing the cloud top height,

cloud base height (retrieved from Micropulse Lidar), and the aircraft G-1 altitude for the July 18, 2017 flight. The cloud extends

from about 2 km to 5 km altitude, with the cloud top and base heights varying over time. The G-1 flew level through different

cloud layers (09:40-10:35 UTC and 11:00 -11:40 UTC). The G-1 also porpoises in and out of the cloud between 10:27 and 10:33 UTC.

Fig. 6 (b) compares the time series of total droplet concentration from three different sources. The blue stars depict the droplet concentration retrieved from the Raman lidar measurements, the maroon dots show the droplet concentration measured by the G-1 cloud probe, and the black stars show the droplet concentration by the G-1 measurements averaged to the lidar retrieval times. Cloud droplet number concentration ($N_{drop, retrieval}$) is derived using the algorithms/methods developed by Snider et al. (2017), which is based on the ground-based Raman lidar particulate extinction profile from the Raman Lidar Vertical Profiles Feature Detection and Extinction (RLPROFFEX) Value Added Product (Chand et al., 2023). The $N_{drop, retrieval}$ achieves good results for stratocumulus cloud while the cloud liquid water content profiles are closer to adiabatic/sudo-adiabatic condition. The lidar retrievals and the G-1 cloud probe measurements agree well, while the lidar retrieved value has a lower droplet concentration near the cloud base between 9:40 and 9:55 UTC and between 11:10 and 11:40 UTC.

A statistical comparison of the total droplet number concentration ($N_{drop}$) between the G-1 measurements and the lidar retrieval shows that the two measurements are highly correlated in the concentration ranges up to 200 cm$^{-3}$, and the G-1 cloud probe measures a slightly higher droplet concentration than the lidar retrieval. Recent studies (Tang et al., 2022; Tang et al., 2023; Zhang et al., 2023b) also pointed out the need for developing appropriate criteria to quantify the cloud remote sensing retrievals better using in situ measurement because the lidar is looking straight up and measuring clouds passing over while the aircraft is flying legs overhead and probing different cloud layers. The large discrepancy might be due to the cloud retrievals capturing the cloud base mainly while the G-1 samples through a portion of the cloud.

### 3.2. Earth system model aerosol-cloud diagnostics package evaluation

An Earth System Model (ESM) aerosol-cloud diagnostics (ESMAC Diags) package (Tang et al., 2022; Tang et al., 2023) has been developed to facilitate the routine evaluation of the DOE's Energy Exascale Earth System Model (E3SM) simulated aerosol, cloud, and aerosol-cloud interactions quantities with the in-situ surface, aircraft, and ship measurements. ESMAC Diags reads in datasets from selected field campaigns and model outputs with some processing and quality controls, then generates a set of diagnostics plots and metrics, such as mean, root-mean-square error and correlation of aerosol and cloud variables. To run the simulation for comparison with aircraft field campaign data, we configure the model according to the

Atmospheric Model Intercomparison Project protocol(Gates et al., 1999), using real-world initial conditions and nudging simulated winds toward Modern-Era Retrospective analysis for Research and Applications (MERRA-2) reanalysis data (Gelaro et al., 2017). Then, we save the hourly output for the field campaign area and utilize an "aircraft simulator" strategy

(Fast et al., 2011) to extract the closest model grid and level-matching aircraft measurements. More details about the model configurations can be found in Tang et al. (2022).

With this diagnostic package, various types of diagnostics and evaluation metrics are performed for aerosol number, size distribution, chemical composition, cloud condensation nuclei (CCN) concentration, and various meteorological quantities to assess how well E3SM represents observed aerosol properties across spatial scales (Tang et al., 2022). Data from two ARM

airborne field campaigns (HI-SCALE and ACE-ENA) have been included in the current version of the ESMAC Diags package. The integrated dataset discussed in this study provides a more consistent input data format than prior data files for the ESMAC Diags. One example is shown in Fig 7. During the intensive operation period (IOP) in the ACE-ENA field campaign (between June 15 to July 20, 2017),  three aerosol number concentrations (for aerosol particles larger than 3 nm, 10 nm and 100 nm) and the cloud condensation nuclei concentration (at 0.1% supersaturation) were compared with the E3SM model (version 2)

simulation using the ESMAC Diags package. Although the E3SM qualitatively reproduced the observation, it overestimated accumulation-mode aerosols and CCN concentration over the ENA regions. In addition, larger discrepancies were observed at the lower altitudes (< 500 m above the ground level) for the aerosol number concentrations (>3 nm and >10 nm), which might be due to the weak representation of nucleation mode aerosol in the model. It did not capture the vertical variation of the CCN concentration, which indicates that process level improvement is still needed for the E3SM over the Atlantic Ocean.

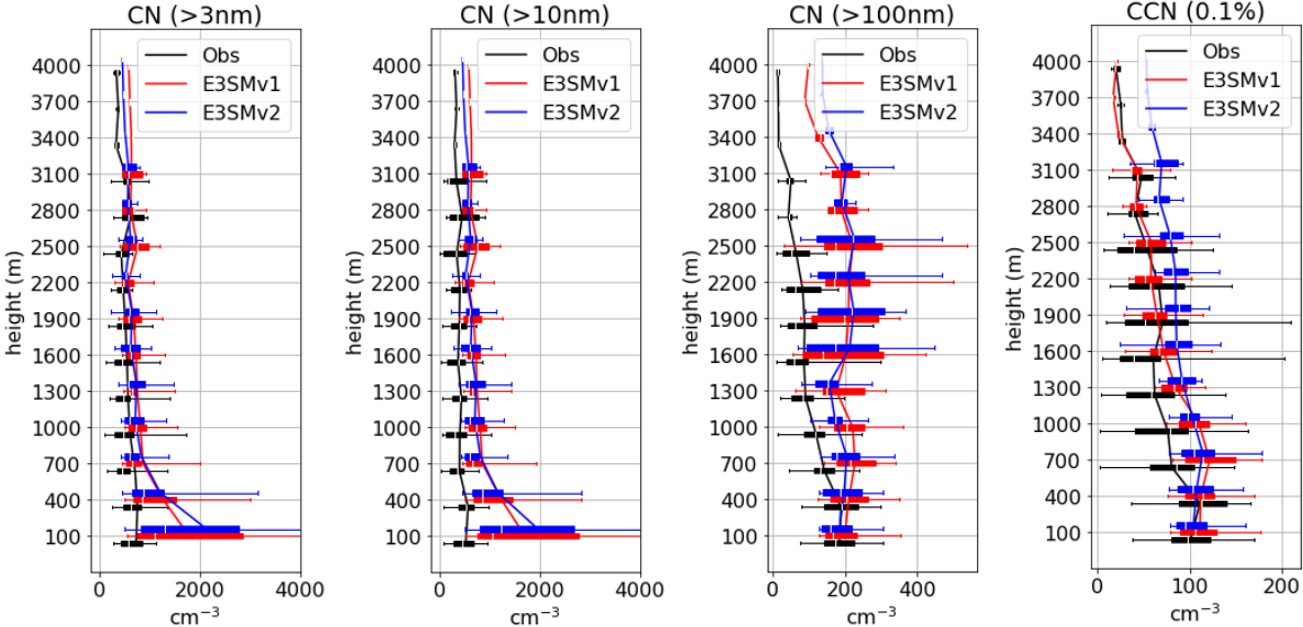

**Figure 7.** An example of E3SM model evaluation using an integrated flight dataset with the ESMAC Diags package (June 15 to July 20, 2017, ACE-ENA field campaign). The percentile box represents the 25th and 75th percentiles, and the bar represents the 5th and 95th percentiles.

### 3.3. Data collection – challenges and future potential.

We summarize the challenges and data collection limitations encountered during the 7 field campaigns in Table 3. The lessons learned from these campaigns suggest that ensuring data quality and enhancing data collection variability require the following key strategies:

- Regular sensor calibration to maintain accuracy with scheduled, impromptu, and well-documented validation methods ensuring reliable measurements.
- Cross-validating and monitoring sensor performance using redundancy, dataset fusion, and statistical techniques to identify inconsistencies and malfunctions.
- Diversified sampling strategies to ensure comprehensive data representation across varying conditions.
- Leveraging model simulations and statistical studies based on previous measurements to refine data collection methods and anticipate potential issues, leading to more effective and targeted data acquisition.

These combined approaches ensure robust data collection, improve measurement accuracy, and provide reliable data products for the community of users.

Future work can expand data collection and merge and facilitate further investigations into atmospheric chemistry, aerosol properties, aerosol-cloud interactions and their representation in Earth System Models. To support future research needs, the

ARM data center plans to work with the AAF instrument mentors and community experts to standardize this merged data product. Future deployments will use a new airborne platform (Challenger 850) and include more baseline airborne measurements (https://arm.gov/capabilities/instruments?type[0]=armobs&category[0]=Airborne%20Observations). For example, we plan to add the liquid water content measurements from the Multi-Element Water Content System (https://arm.gov/capabilities/instruments/wcm-air) and solar radiation measurements from multifilter radiometers (https://arm.gov/capabilities/instruments/mfr-air) into the future merged dataset.

In addition to the baseline measurements, we plan to offer data integration options through the ARM Data Integrator tool (details in section 4), allowing campaign principal investigators or community users to flexibly incorporate additional data into the merged data product. This approach provides users with the ability to target specific science themes. For example, one proposed data product is to include additional aerosol optical properties with this AAF merged dataset. The atmospheric community can expand research on the evolution of aerosol particles from wildfires, particularly on how different combustion phases (flaming vs. smoldering) result in varying chemical compositions and quantities of emitted aerosols with additional chemical composition and gaseous phase concentration data. This custom-built merged dataset is crucial for improving air quality models and understanding the climate impacts of biomass burning. Continuing to investigate the influence of urban pollution on natural aerosol formation, similar to the studies conducted in the Amazon during GoAmazon2014/5, will provide new insights into aerosol composition. Potential data for such a study could be non-airborne remote-sensing data. Combining airborne data with ground-based remote sensing data allows the exploration of interactions between different aerosol types and evolving cloud and precipitation patterns. Quantifying these interactions can improve models and understanding, aiding the development of strategies for mitigating anthropogenic impacts on natural environments.

By strategically combining long-term ground-based remote sensing measurements with high-resolution airborne data, researchers can achieve more robust analyses of atmospheric processes, leading to more accurate scientific findings and better-constrained models. For instance, ground-based sensors can continuously monitor a specific location, while targeted airborne missions can capture critical in situ measurements during specific events that are not retrievable by remote sensing to better study cloud evolution, pollutant transport, or extreme weather.

Table 3. Challenges and data collection limitations of the seven AAF field campaigns between 2013-2018.

| AAF-supported field campaigns | Challenges and data collection limitations |
|---|---|
| Biomass Burning Observation Project (BBOP), 2013 | <ul><li>Air traffic control regulations and safety concerns limited the ability to sample immediately after emissions, focusing instead on measurements 15 minutes to several hours post-emission.</li><li>Due to the high aerosol number concentrations in the fire plumes, most instruments were operated near their upper limits. Post-processing was implemented to minimize the coincidence problem with the particle counting but couldn't entirely eliminate the issue.</li></ul> |

| | |
|---|---|
| Observations and Modeling of the Green Ocean Amazon (GoAmazon), 2014 | • The Amazon rainforest experiences frequent cloud cover and heavy rainfall, which significantly interfered with flight planning and the number of viable data collection periods.<br>• The Amazon's high humidity and frequent rainfall can negatively affect airborne instruments and sensors. Thus, a dryer system was integrated into the inlet system to ensure the aerosol data collection under dry conditions. |
| ARM Cloud Aerosol Precipitation Experiment (ACAPEX), 2015 | • Few ARs made landfall in northern California during the campaign, and many days had clear skies, which limited opportunities for studying aerosol-cloud-precipitation interactions.<br>• While flying through heavy precipitation, water accumulation in cloud probes posed difficulties for accurate measurement collection. Some archived data had to be flagged during the post-processing. |
| Airborne Carbon Measurements (ACME V), 2015 | • Dense fog and low-elevation clouds in September limited flight operations over coastal sites, affecting data collection during this period.<br>• Aerosol measurements were limited due to deployment and budget restrictions in the Arctic environment during this campaign. |
| Holistic Interactions of Shallow Clouds, Aerosols, and Land-Ecosystems (HI-SCALE), 2016 | • The G-1 aircraft operations were based at a location 150 km away from the ARM SGP central facility, which resulted in extra flight hours during transit.<br>• The hot weather posed an additional challenge, reducing flight time and affecting the optimal flight times during the day.<br>• Another challenge was the aircraft's payload capacity, a common issue for all research aircraft, as we aimed to carry more instruments than the aircraft could accommodate. As a result, we had to sacrifice some optical measurements to stay within the payload limits. |
| Aerosol and Cloud Experiments in the Eastern North Atlantic (ACE-ENA), 2017/8 | • The Eastern North Atlantic frequently experiences rough weather, including high winds, storms, and turbulence, posing significant challenges for flight operations.<br>• Occasional impact of anthropogenic emissions from Graciosa Island on G1 measurements, especially at the lowest sampling altitudes.<br>• Terrain blockage limited the coordination of scanning cloud radar operation and G1 sampling when the wind is in certain directions (One major feature of ACE-ENA is the synergy between G1 measurements and observations at the ENA site).<br>• The air space access is restricted near the ground (ENA) site due to incoming and outgoing commercial flights. |
| Cloud, Aerosol, and Complex Terrain | • The intricate interactions between the boundary layer, orographic, low-level jet, and frontal circulations produced tremendous variability in aerosol and cloud conditions. |

| Interactions (CACTI), 2018 | • Some other challenges included (1) the safety risk from intense, quickly evolving storms in the vicinity of the flying aircraft and hangar, (2) clouds intersecting or being close to high terrain, making it impossible to sample the cloud base; (3) power outages, which raised concerns about INP filters thawing during the storage, and (4) weather warnings such as icing conditions. |
|---|---|

## 4. Data availability and code availability

The ARM Data management system treats all data from the G-1 airborne deployments as field campaign data streams, meaning the data were collected during the intensive observational periods rather than over long-term observation. The data (https://www.doi.org/10.5439/1999133) in this manuscript were produced following ARM data file standards and archived through the ARM data ingest process (Prakash et al., 2016) under the Creative Commons License. In 2016, the ARM G-1 raw instrument data was directly "ingested" into netCDF format and archived automatically during and after flight operations. To

provide a dataset with a uniform data format, the G-1 payload scientists (ARM mentors) reprocessed data for field campaigns between 2013 and 2015 and converted the historical data from ICARTT to netCDF. Note that a specific directory has been created for reviewers to access the data at https://adc.arm.gov/essd/.

The ARM data system uses a multi-tiered data processing approach (Prakash et al., 2016) that iteratively processes the instrument data to produce higher-level data products. Data is first processed from the instrument's raw data format (data level

00) to netCDF format. During this initial processing conversion of geophysical units and application of calibration factors is performed as appropriate (documented in a and b level data products). Quality controls can also be applied (creating b1 level data files).   Additional processing can be added to further increase the level of these files with 'higher value' to store as a data level 'c1'. For example, mentor-edited data files with additional quality improvement calibration are usually considered the 'c' level data product.

The content of ARM data files is structured in three main sections: dimensions, variables, and global attributes. As time-series measurements, the time dimension of ARM products is considered 'unlimited'. Per netCDF3 requirements, the unlimited dimension 'time' is the first dimension of a variable that uses the time dimension. The variables encompass coordinate variables reporting dimension values, primary measurements (recommended for scientific use), supporting measurements (e.g., diagnostics and quality), and location variables detailing latitude, longitude, and altitude. Variables are equipped with

supporting attributes to facilitate the user's understanding and interpretation. These include a "long_name" for unique descriptions, "units" conforming to unit conventions, and a "missing_value" to represent no data. A "standard_name" attribute, following the Climate Forecast (CF) standard, is assigned when applicable. The final section in ARM netCDF files consists of global attributes containing information related to the platform's location, time interval, calibration procedures (if available), and contact information for instrument mentors or principal investigators.

A final merged product (aafmerged.c1) was created to provide users with all G-1 airborne measurements in a single file. This merged data product is produced using the ARM Data Integrator (ADI, https://github.com/ARM-DOE/ADI), a framework designed to automate data retrieval, integration, and the creation of time-series NetCDF data products. ADI allows users to seamlessly combine data products, extract specific variables, and transform them into user-defined coordinate systems. The time dimension of the merged data product aligns with the input aafnaviwg.c1 datastream. All other instrument data was

mapped onto this sampling period using ADI's nearest-neighbor transformation method.

All data products produced by the ARM data system (Prakash et al., 2016) adhere to ARM Data Standards and are made available to the user community via the ARM Data Center in files using a naming convention (detailed in Table 4) of (sss)(inst)(qualifier)(temporal)(Fn).(dl).(yyyymmdd).(hhmmss).nc.

Table 4. The naming convention of the merged data product.

| Names | Variable information |
|---|---|
| sss | the three-letter ARM site identifier (e.g., ena for the ACE-ENA field campaign data). |
| inst | the ARM instrument abbreviation (e.g., aafcpcf), or the name of an ARM Value Added Product (VAP), such as aaf for this integrated multi-sensor dataset. |
| qualifier | an optional qualifier that distinguishes multiple data from similar instruments but on different platforms. |
| temporal | an optional description of temporal data resolution (e.g., 1s). |
| Fn | the two- or three-character ARM facility designation (e.g., F1 for the G-1 aircraft). |
| dl | the two-character descriptor of the data level, consisting of one lower-case letter followed by one number (e. g. c1). |
| yyyymmdd &hhmmss | the coordinated universal time (UTC) date and time indicates the start time of the first data point measured. |
| nc | the netCDF file extension. |


For example, a netCDF file produced for the G-1 airborne deployment at the ACE-ENA field campaign that includes quality-controlled data (in geophysical units) collected starting at 08:31:45 UTC on July 18, 2017, is named as enaaafF1.b1.20170718.083145.nc.

All AAF data collected and ingested after 2016 aligns with the ARM standard data process and format described above. Due to the complex nature of various airborne measurements, some G1 instrument data has been edited by the mentor (such as the cloud probe) to add additional value. These mentor-edited data are directly ingested from raw data into "c1" level data products. For the historical campaign data before 2016, we reprocessed the AAF data following the standard process if the raw data were in a similar format. Otherwise, the mentor-edited data were used as the input variables to the ADI process to create the

aafmerged.c1 products. The details about individual data included in the final merged data file for each field campaign are

listed in Table S2, as are the primary measurements and their associated standard names used in the recommended aafmerged.c1 product. A "standard_name" is listed in Table S4 for the primary variables, which is consistent with the naming convention based on the Climate Forecast (CF) standard.

## 5. Summary

This paper provides an overview of the platform, the aircraft instrumentation, flight tracks, and data collected during the ARM airborne field campaigns and introduces information on data quality control. While numerous studies based on AAF data have both directly and indirectly demonstrated the quality of the datasets, this paper further reinforces this by providing specific examples. It compares in situ measurements with other collocated observations, offering additional evidence to underscore the reliability of the AAF data. A merged dataset containing each flight's meteorological, aerosol and cloud information was

generated for seven AAF field campaigns between 2013 and 2018. The data from 766.4 hours of research flights were collected over multiple continents and in various environmental conditions.

Four of the seven field campaigns were based in the U.S. One campaign collected data from the wildfires in the U.S. Pacific Northwest and agricultural burns in the lower Mississippi River valley as part of the BBOP in 2013. In 2015, the ARM Cloud Aerosol Precipitation Experiment provided data on atmospheric rivers and associated aerosol-cloud interactions that produce

heavy precipitation on the U.S. West Coast during the early spring. Research data from ACME V, collected during the summer of 2015, gave scientists insight into trends and variability of trace gases in the atmosphere over the North Slope of Alaska to improve Arctic climate models. In the early summer and autumn of 2016, HI-SCALE provided an extensive dataset geared toward coupled processes that affect the life cycle of shallow clouds through the interaction among aerosol, cloud, land surface and ecosystems.

In 2014 (March and October), the airborne sampling moved outside of the U.S. to Manus in central Amazonia, Brazil, where residential and industrial emissions were extensively characterized by flights of the G-1. The GoAmazon2014/15 aircraft campaign data are being integrated with aquatic and terrestrial ecosystem measurements to quantify anthropogenic perturbations to a usually pristine tropical environment. Another international airborne mission was carried out in the Eastern North Atlantic region. The ACE-ENA campaign saw the G-1 aircraft fly from Terceira Island in the Azores during the summer

2017 and winter of 2018. The campaign studied both seasons to measure key aerosol and cloud processes under various meteorological and cloud conditions with different aerosol sources. Then the G-1 deployed to the Sierras de Córdoba range in central Argentina from October to November 2018 to study orographic convective cloud interactions with their surrounding environment.

The combined observational data from these field campaigns facilitates studying of atmospheric processes, such as boundary

layer processes, aerosol-cloud-precipitation interactions, and land-atmosphere-cloud interactions across a wide range of conditions. Although each field campaign faced different challenges and data collection limitations, many previous studies have benefited from the G-1 field campaign data (Gu et al., 2017; Creamean et al., 2018; Fast et al., 2019a; Shrivastava et al.,

2019; Berg et al., 2020; Yeom et al., 2021; Zhang et al., 2021; Wang et al., 2023b; Wang et al., 2023a; Zhang et al., 2023b), and more manageable data access would support new users in the research community further. By incorporating data from multiple sources, these ARM datasets and open-source tools can provide more accurate and reliable information and assist the model simulation/prediction improvement. Overall, a merged airborne aerosol, cloud, and trace gas dataset covering seven field campaigns is a powerful tool for atmospheric scientists, supporting a more comprehensive understanding of atmosphere processes impacting the climate. We hope our efforts will encourage broader usage of the ARM data and enhance the collaboration between the ARM user facility and the atmospheric science community.

## Author Contributions.

FM led the formulation of this paper. FM and ST provided the figures. JMT, FM, MSP, JES, JMC, DZ and BS participated in the data collection and preparation of the field campaigns. JDF, ACV, JW, SM, SCB, LRL, and LK led the field campaigns and provided revision comments to the manuscript. BDE, SKG and KWB worked on the data formatting and merging processes. FM wrote the draft and all coauthors provided editing suggestions to the manuscript.

## Competing Interests.

The authors declare that they have no conflict of interest.

## Metadata Information.

The ARM includes sufficient metadata in each netCDF file to facilitate the user's understanding and interpretation. Public Data Usage Rights: This work is licensed under the Creative Commons Attribution 4.0 International License. To view a copy of this license, visit https://creativecommons.org/licenses/by/4.0/.

## Acknowledgements.

This work has been supported by the Office of Biological and Environmental Research (OBER) of the US Department of Energy (DOE) as part of the Atmospheric Radiation Measurement (ARM) User Facility and the Atmospheric System Research (ASR) program. ST and JDF were also supported by the Enabling Aerosol-cloud interactions at GLobal convection-permitting scalES (EAGLES) project (74358), funded by the U.S. Department of Energy, Office of Science, Office of Biological and Environmental Research, Earth System Model Development (ESMD) program area. Battelle operates the Pacific Northwest National Laboratory (PNNL) for the DOE under contract DE-AC05-76RL0 1830. This work benefited from the assistance of OpenAI's ChatGPT for refining the language used in this document.

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
