# Peer review of "Atmospheric Radiation Measurement (ARM) airborne field campaign data products between 2013 and 2018"

_Earth System Science Data, 2024_

## Author Comment (AC2)

**Review 2 comment on essd 2024-97**

The manuscript gives an overview of the detailed data collected during those seven field campaigns by the ARM program and how airborne measurements catch the atmosphere's detailed atmospheric processes. A data set integrated from these campaigns is an important resource for the study of various atmospheric phenomena but more so for aerosols, clouds, and trace gases. The authors have indeed succeeded in compiling, standardizing, and making these datasets available, which will support ongoing and future research in our community. Overall, this manuscript is well-organized and of clear scientific significance. I want to give him some advice:

We sincerely appreciate the reviewer's comments and suggestions. We have provided responses below and revisions to the manuscript to address each comment.

1. The manuscript may benefit from more discussions of the limitations and uncertainties associated with the data, particularly regarding the comparison between airborne and ground-based measurements. While the paper touches on the challenges of comparing these two types of measurements, a more thorough exploration of the potential biases introduced by differences in spatial and temporal resolution would help us understand more about the dataset.

**Response:** Thank you for this constructive advice. We have added Table S3 concerning uncertainties associated with the data. Regarding the comparison between airborne and ground-based measurements, we revised section 3 to the below paragraphs.

" The AAF's airborne measurements are considered accurate and reliable because they are obtained directly from the atmosphere using well-calibrated instruments. Numerous prior studies have systematically assessed ARM data quality, employing methods such as laboratory evaluation based on community-accepted standards, comparing similar properties across different instruments, and conducting intercomparisons across diverse platforms (from ground to airborne or airborne to airborne) (Bond et al., 1999; Lance et al., 2010; Kassianov et al., 2015; Kassianov et al., 2018; Mei et al., 2020; Zawadowicz et al., 2021; Kulkarni et al., 2023; Tang et al., 2023; Zhang et al., 2023b). Table S3 lists AAF measurements and uncertainties of atmospheric properties, including temperature, humidity, aerosol concentrations, cloud particle sizes, and radiation levels. (Mei et al., 2020) "

" This study further demonstrates the comparison of AAF data with ground-based remote sensing retrieval. The ARM G-1 aircraft was deployed above or near the ENA and SGP sites during field campaigns like ACE-ENA and HI-SCALE. Various instruments employing diverse observational techniques have measured atmospheric parameters, aerosol, and cloud properties from ground and airborne perspectives. These coordinated deployments enable a thorough assessment of robustness and statistical representativeness across collocated measurements.

Comparing airborne and ground-based measurements involves evaluating data from two platforms that differ in spatial and temporal resolutions, and measurement techniques. Thus, three potential biases exist in the measurements – spatial, temporal, and instrumental. The instrumental bias is typically due to the differences in sensors, calibration, and data processing techniques between the two platforms. Airborne measurements usually provide in situ spatiotemporal data over leveled flight legs at different altitudes and can capture data over various, even difficult-to-access, terrains. Meanwhile, ground-based remote sensing data usually provide continuous monitoring at a fixed location with limited spatial coverage or less vertical resolution. Our efforts focus on minimizing the temporal and spatial biases to ensure accurate and meaningful comparisons. We selected the comparison period by aligning the data acquisition times for both airborne and ground-based measurements as closely as possible. To ensure that both airborne and ground-based measurements are georeferenced accurately. For instance, ground-based remote sensing uses height or the altitude above the ground level (AGL) as the vertical geographic coordinate. In contrast, airborne data usually uses the mean sea level (MSL) altitude, which can be converted to the AGL. We then use interpolation techniques to match the spatial resolutions of airborne and ground-based data. "

2. The discussion of data quality could also include the differences and specific challenges between different field campaigns. For instance, variations in environmental conditions across different campaigns may have unique limitations in data collection and processing.

**Response:** Thank you very much for this suggestion. We have added section 3.3 "Data collection – challenges and future potential" to share lessons learned from each campaign in Table 3.

3. The manuscript adds more case studies or examples of how the dataset has been or could be used in specific research applications, which would provide practical context for its utility in past research.

**Response:** Thank you so much for this suggestion. We have provided more case studies and examples based on past research in section 3.

" The AAF airborne data are also often used as a benchmark or standard for other measurements, especially those from remote sensing technologies such as satellites, ground-based radars, and lidars. Junghenn Noyes et al. validated remote sensing retrievals with help from ground-based and airborne measurements. Their study enhanced the understanding of smoke particle behavior and its implications for remote sensing. (Junghenn Noyes et al., 2020) Mech et al. showcased how integrating airborne data into the PAMTRA (Passive and Active Microwave TRAnsfer) validation process enhances the model's skill in accurately simulating microwave measurements. The detailed comparison between simulated and observed data helps understand the model's performance in real-world conditions, leading to a more robust and reliable tool for atmospheric research. (Mech et al., 2020) Yang et al. developed a new method to estimate supersaturation fluctuations in stratocumulus clouds using ground-based remote-sensing retrievals. Then, they

used the airborne data to validate these estimations. (Yang et al., 2019) Wu et al. retrieved profiles of marine boundary layer (MBL) cloud and drizzle microphysical properties from ground-based observations, validated by aircraft measurements over the Azores. (Wu et al., 2020) Zhang et al. (2023) evaluated cloud droplet number concentrations using multiple ground-based methods validated through aircraft in situ measurements. (Zhang et al., 2023a)

Furthermore, research based on collected aircraft data led to advancements, characterization, and understanding of atmospheric processes. (Martin et al., 2017; Fast et al., 2011; Fast et al., 2019a; Fast et al., 2019b; Varble et al., 2021; Wang et al., 2022) For example, various studies have utilized aircraft measurements to characterize aerosol and cloud properties while advancing the understanding of aerosol chemistry and cloud microphysical properties and processes, including investigations over the North Atlantic, Amazon basin, and Southern Great Plains. (Shrivastava et al., 2019; Shilling et al., 2018; Zawadowicz et al., 2020; Wang et al., 2023b; Fast et al., 2024) The airborne data has also been used to examine the vertical variability of aerosol properties over the Southern Great Plains, contributing to a better understanding of the distribution and impact of aerosols at different atmospheric levels. (Wang et al., 2016; Fast et al., 2022) "

4. The manuscript may discuss the potential for future updates or expansions of the dataset, including the incorporation of additional variables or the development of new analytical tools to enhance its utility.

**Response:** We appreciate this advice and added section 3.3 "Data collection – challenges and future potential" to discuss the potential for future updates or expansions based on the science community needs.

[revised manuscript text omitted]

---

## Author Response (AR2)

Response to Figure S3 questions.

Thank you very much for your detailed explanation.

We generated the figure by overlaying our own data with the map library from Python. DOE ARM Data Center has implemented this quicklook plot with its data visualization tool under https://dq.arm.gov/dq-zoom/. We have updated the figure caption to the below. Please take a look and let us know if we need to modify further.

[Figure]

Figure S3. An example quicklook plot using ARM Atmospheric Data Community Toolkit developed using Python (https://github.com/ARM-DOE/ACT).